# Single Nucleotide Polymorphisms in Amlodipine-Associated Genes and Their Correlation with Blood Pressure Control among South African Adults with Hypertension

**DOI:** 10.3390/genes13081394

**Published:** 2022-08-05

**Authors:** Charity Masilela, Oladele Vincent Adeniyi, Mongi Benjeddou

**Affiliations:** 1Department of Biochemistry, North West University, Mahikeng 2735, South Africa; 2Department of Family Medicine, Walter Sisulu University, East London 5200, South Africa; 3Department of Biotechnology, Faculty of Natural Science, University of the Western Cape, Cape Town 7535, South Africa

**Keywords:** amlodipine, single nucleotide polymorphisms, South Africa, uncontrolled hypertension

## Abstract

Objective: This study describes the single nucleotide polymorphisms (SNPs) in amlodipine-associated genes and assesses their correlation with blood pressure control among South African adults with hypertension. Methods: In total, 304 hypertensive patients on amlodipine treatment belonging to the indigenous Swati, Xhosa and Zulu population groups of South Africa were recruited between June 2017 and June 2019. Participants were categorized into: controlled (blood pressure < 140/90 mmHg) and uncontrolled (blood pressure ≥ 140/90 mmHg) hypertension. Thirteen SNPs in amlodipine pharmacogenes with a high PharmGKB evidence base were selected and genotyped using MassArray (Agena Bioscience^TM^). Logistic regression was fitted to identify significant associations between the SNPs and blood pressure control with amlodipine. Results: The majority of the participants were females (76.6%), older than 45 years (89.1%) and had uncontrolled hypertension (52.3%). Of the 13 SNPs genotyped, five SNPs, rs1042713 (minor allele frequency = 45.9%), rs10494366 (minor allele frequency = 35.3%), rs2239050 (minor allele frequency = 28.7%), rs2246709 (minor allele frequency = 51.6%) and rs4291 (minor allele frequency = 34.4%), were detected among the Xhosa participants, while none were detected among the Swati and Zulu tribal groups. Variants rs1042713 and rs10494366 demonstrated an expression frequency of 97.5% and 79.5%, respectively. Variant TA genotype of rs4291 was significantly associated with uncontrolled hypertension. No association was established between blood pressure response to amlodipine and the remaining four SNPs. Conclusions: This study reports the discovery of five SNPs in amlodipine genes (rs2239050, rs2246709, rs4291, rs1042713 and rs10494366) among the indigenous Xhosa-speaking tribe of South Africa. In addition, the TA genotype of rs4291 was associated with blood pressure control in this cohort. These findings might open doors for more pharmacogenomic studies, which could inform innovations to personalised anti-hypertensive treatment in the ethnically diverse population of South Africa.

## 1. Introduction

Hypertension is one of the most potent cardiovascular risk factors that affects over a billion people worldwide [1]. The 2016 Demographic and Health Survey detected a prevalence for arterial hypertension of up to 23% (women) and 13% (men) among South Africans adults [2], whereas newer studies reported a national prevalence of up to 60% [1]. Additionally, studies conducted in KwaZulu Natal, Eastern Cape, Gauteng and the Limpopo province of South Africa have reported a range of prevalence between 12.4% and 52.0% [1,3,4,5]. Arterial hypertension is highly prevalent among South African adults and, unfortunately, the rate of uncontrolled hypertension is high as well [6]. A recent study reported an overall prevalence of uncontrolled hypertension of 56.83% among rural dwellers in the Mpumalanga province [7], whilst a 75.5% and 51.0% prevalence was reported in the rural Eastern Cape and KwaZulu Natal, respectively [6,8]. As hypertension care is slowly evolving to include behavioural and socio-demographic factors, a large body of evidence suggests that utilizing genetic factors to characterise response to pharmacological treatments offers a new approach to tailoring anti-hypertensive treatment to patients [9,10]. Consequently, there is a need for robust anti-hypertensive management strategies that will utilize genetic factors during both drug selection and dosing to ensure optimal blood pressure control.

Amlodipine is a long-acting third-generation calcium channel blocker (CCB) that has been shown to effectively lower blood pressure and reduce cardiovascular disease risk among hypertensive patients [11,12]. However, blood pressure response to amlodipine is highly variable and several studies have investigated the potential genetic polymorphisms that could account for the inter-individual variability observed across individual patients and populations [13]. These studies examined single nucleotide polymorphisms (SNPs) that occur in genes that are directly involved in the pharmacodynamics and pharmacokinetics of amlodipine, such as the voltage-gated calcium channel α1C (CACNA1C) [14,15]. The CACNA1C gene encodes for an α-1 subunit of a voltage-dependent calcium channel that mediates the influx of calcium ions into the cell upon membrane depolarization [16]. This gene harbors SNPs (rs2239050, rs2238032 and rs527974) that have been implicated in hypertension. Both rs2238032 and rs2239050 were associated with uncontrolled hypertension among Caucasian patients [13,15]. On the contrary, Japanese carriers of the promoter variant rs527974 with uncontrolled hypertension showed increased amlodipine sensitivity [13]. Further, amlodipine is largely metabolised in the liver by the enzyme cytochrome P450 3A5, which is encoded by the gene Cytochrome P450 Family 3 Subfamily A Member 5 (CYP3A5) [17]. The literature suggests that Chinese carriers of CYP3A5*3/*3, CYP3A5*3 and CYP3A5*6 polymorphisms demonstrate increased amlodipine metabolism, as well as increased CYP3A enzyme efficacy [18].

In addition, SNPs that are harbored by genes that are indirectly implicated in the pharmacokinetics of the anti-hypertensive effect of amlodipine, such as angiotensin converting enzyme (ACE) and angiotensinogen (AGT), have been examined [9,13]. Both ACE and AGT are central components of the renin–angiotensin system (RAS) that controls blood pressure by regulating the volume of fluids in the body [19]. Variant rs4291 of the ACE gene was strongly associated with the incidence of high blood pressure [9]; however, its direct association with blood pressure response to amlodipine remains elusive. Similarly, African American carriers of the minor allele of rs11122576 from the AGT gene who were undergoing amlodipine therapy showed a decreased risk of coronary heart diseases; however, no clear association with blood pressure in response to amlodipine has been established [20]. Additionally, β-adrenergic receptor (ADRB2) and nitric oxide synthase-1-adaptor protein (NOS1AP) are part of the sympathetic and para-sympathetic nervous systems and are known to be involved in the pathophysiology of hypertension [21,22]. The minor allele G of rs10494366 of the NOS1AP gene was associated with an increase in cardiovascular mortality among Caucasian users of amlodipine [22]. Furthermore, patients with the AA genotype of rs1042713 (ADRB2) demonstrate poor efficacy of cardiovascular drugs, including ACE inhibitors [23]. Nevertheless, no direct association with blood pressure response to amlodipine has been established for both SNPs. More studies need to be conducted to establish a clear association between these SNPs and blood pressure response to amlodipine.

The discovery of SNPs facilitated patient stratification for many diseases, including hypertension, and allowed physicians to adopt pharmacogenomics-based approaches in the selection of anti-hypertensive drugs to improve drug efficacy among patients [24]. Although these findings have added relevant biological insights regarding disease phenotypes, only a small fraction of published data have focused on indigenous South African populations and the genetic contribution of SNPs on therapeutic drug response, this is particularly evident in the case of amlodipine. Moreover, the lack of African-specific genomic data has slowed down our understanding of the underlying mechanisms implicated in blood pressure response to amlodipine and the generation of new leads relevant in personalised anti-hypertensive treatment for local populations. To improve the treatment of hypertension in South Africa, studies on its ethnically diverse sub-populations are needed. The current study describes the single nucleotide polymorphisms (SNPs) in amlodipine-associated genes and further assesses their correlation with blood pressure control among South African adults with hypertension.

## 2. Methods

### 2.1. Ethical Approval

The study protocol was approved by the Senate Research Committee of the University of the Western Cape (Ethics approval number: BM/16/5/19). Permission to implement the study protocol was granted by the clinical governance of the respective hospitals in the Eastern Cape and Mpumalanga Provinces. Participants were issued with a research information sheet detailing the purpose and process of the study; it was made available in various indigenous languages (Swati, Xhosa and Zulu). Each participant signed an informed consent form as evidence of voluntary participation in the study. Participants’ rights to privacy and confidentiality of medical information were respected during and after the study.

### 2.2. Patient Selection and Data Collection

In total, 304 patients with hypertension belonging to the indigenous Nguni (Swati, Xhosa and Zulu) population groups of South Africa were recruited from Cecilia Makiwane Hospital (East London, Eastern Cape), Piet Retief Hospital, Thandukukhanya Community Health Center and Mkhondo Town Clinic (Mkhondo, Mpumalanga) between June 2017 and June 2019. Participants were eligible if they were 18 years or older and were on continuous anti-hypertensive therapy treatment for at least a year. Individuals who were bedridden, pregnant and clinically unstable were excluded from the study.

A trained research nurse measured the blood pressure (BP) of each participant using a validated automated digital blood pressure monitor (Macrolife BP A 100 Plus model) according to standard protocols. The BP was recorded in triplicate and the average was used for analysis. Patients were categorized as: controlled (blood pressure < 140/90 mmHg) and uncontrolled (blood pressure ≥ 140/90 mmHg). DNA samples were collected in the form of buccal swabs and stored at −20 °C.

In addition, the nurse also measured the weight of each participant to the nearest 0.1 kg using a digital scale (Tanita-HD 309, Creative Health Products, Ann Arbor, MI, USA) and height to the nearest of 0.1 cm using a mounted stadiometer. Participants wore minimal clothing and no shoes. Body Mass Index (BMI) was estimated as weight (kg) divided by height in squared meters (m^2^) and was categorized based on WHO criteria as underweight, normal, overweight and obese (30 or greater kg/m^2^). The age, ethnicity, smoking status and salt intake were self-reported by each participant and documented in a proforma designed for this study. Prescribed drugs for each participant were retrieved from their medical records. Physical activity was categorised as active if participants engaged in exercise leading to an increase in heart and respiratory rate, such as gardening (1–3 times per week), or inactive if they did not engage in any physical activity. Smoking status was categorised as never smoked or ever smoked (current smokers or have a history of tobacco use). Salt intake was determined by the survey question do you add salt on the table while eating or on purchased take-away food? Participants who answered “yes” were placed in the increased salt intake category; participants who answered “no” were placed in the low–moderate salt intake category. Prescribed anti-hypertensive drugs included amlodipine alone or in combination with hydrochlorothiazide, enalapril and atenolol.

### 2.3. Laboratory Assessments

Total cholesterol, triglycerides, low-density lipoprotein and high-density lipoprotein cholesterol were assayed from venous blood samples after a minimum of eight-hour fasting by the participants. Lipid profile analysis was conducted by the National Health Laboratory Services (NHLS) of Cecilia Makiwane, Piet Retief and Ermelo Provincial hospitals in accordance with standard protocols and categorised according to the guidelines of The Society for Endocrinology, Metabolism and Diabetes of South Africa (SEMDSA) [25].

### 2.4. DNA Isolation

Genomic DNA was extracted from buccal swab samples using a standard salt-lysis procedure [26]. Briefly, DNA samples were incubated in lysis buffer at 62 °C overnight. Thereafter, DNA was precipitated with NaCl followed by the addition of absolute ethanol and incubated at −80 °C for 30 min. Precipitated DNA was purified using 70% ethanol and resuspended in nuclease-free water. Samples were stored at −20 °C until further use. DNA quantification was conducted using a NanoDrop™ 2000/2000c Spectrophotometer (Thermo Scientific™, Waltham, MA, USA) and Gel Doc™ EZ Gel Documentation System (BIO-RAD, Hercules, CA, USA).

### 2.5. Selection of Pharmacogenomics Biomarkers

Thirteen single nucleotide polymorphisms associated with blood pressure response to amlodipine were selected using the Pharmacogenomics knowledge base [27], Ensembl [28], as well as an extensive survey of recent literature. We focused on genes in pathways directly or indirectly involved in the blood pressure lowering mechanism of amlodipine exhibiting Pharmagkb evidence rating of at least 3 (Table 1).

### 2.6. Genotyping

Two multiplex MassARRAY systems (Agena Bioscience^TM^) were designed and optimised by Inqaba Biotechnical Industries (Pretoria, South Africa) and used for the genotyping of the selected SNPs. The genotyping assay is based on a locus-specific PCR reaction that is followed by a single base extension using the mass-modified dideoxynucleotide terminators of an oligonucleotide primer, which anneals immediately upstream of the site of mutation. The SNP of interest is identified using MALDI-TOF mass spectrometry.

### 2.7. Statistical Analysis

Statistical analyses were performed using Medcalc version 2.2.0.0. The general characteristics of the participants were summarised by using simple descriptive statistics. Associations between alleles, genotypes and blood pressure response to amlodipine were measured using unadjusted and adjusted odds ratios (ORs), 95% confidence interval (95% CI) and *p*-value derived from unconditional logistic regression. In the final model of the adjusted logistic regression analysis, we included rs2239050, rs2246709, rs4291, rs1042713 and rs10494366, where drug regime was used to adjust each SNP. Results for the unadjusted logistic regression model analysis were expressed as crude odds ratios (CORs) and adjusted odds ratios (AORs) for the adjusted logistic regression model analysis. A *p*-value less than 0.05 was considered statistically significant. Bonferroni corrected *p*-values were set at 0.025. Minor allele frequency (MAF) and Hardy–Weinberg equilibrium (HWE) tests were calculated for all the SNPs using Genetic Analysis in Excel (GenAIEx) Version 6.5. (New Brunswick, NJ, USA).

## 3. Results

### 3.1. Baseline Characteristics of the Participants

A total of 304 patients participated in the study, of which 23.3% (*n* = 71) were males and 76.6% (*n* = 233) were females. Most of the study participants were older than 45 years (*n* = 271), 85.1% (*n* = 259) had never smoked, 78.0% (*n* = 237) had low–moderate salt intake and 53.3% (*n* = 159) had blood pressure ≥ 140/90 mmHg. Overall, 4.28% (*n* = 13) of the participants were on amlodipine monotherapy (Table 2).

### 3.2. Descriptive Patterns of Single Nucleotide Polymorphisms Associated with Amlodipine

Thirteen SNPs were selected, and their expression patterns were assessed across three indigenous South African populations (Swati, Xhosa and Zulu). Out of thirteen SNPs, only five were detected among the Xhosa tribe. However, the variant alleles of the five SNPs were not detected among Swati and Zulu participants. rs2239050, rs2246709 and rs4291 were observed among all Xhosa participants (*n* = 122). rs1042713 was observed among 79.5% (*n* = 97) of Xhosa participants, of whom 76.8% (*n* = 63) were females and 78.2% (*n* = 43) were aged between 55 and 65 years. Further, rs10494366 was detected among 97.6% (*n* = 119) of the Xhosa participants, 97.6% (*n* = 80) were females and 98.2% (*n* = 54) aged between 55 and 65 years (Table 3).

The allelic distribution of the five SNPs did not deviate from Hardy–Weinberg equilibrium in the study cohort. The minor allele frequency observed for the selected SNPs in the Xhosa population was compared to world populations listed on Ensambl, that is, in the Luhya people of Kenya, the Yoruba of Nigeria, and African American, Mexican, British and South Asian populations. The SNPs rs1042713 (45.9%), rs10494366 (35.3%) and rs2239050 (28.7%) showed lower minor allele frequency in the Xhosa population in comparison to all selected populations. Variant rs2246709 showed a slightly higher minor allele frequency in the Xhosa population (51.6%) when compared to selected world populations. In comparison to the Yoruba (22.7%) and Luhya people (23.2%), the Xhosa population (34.4%) showed a higher minor allele frequency; however, this was lower than British (41.4%), Mexican (44.0%) and South Asian (38.4%) populations (Table 4).

### 3.3. Association between SNPs and Blood Pressure Control with Amlodipine

In the multivariate (crude) logistic regression model analysis, the genotype TA (rs4291) was independently and significantly associated with uncontrolled hypertension with amlodipine treatment. After adjusting for other factors in the logistic regression model analysis, the magnitude and direction of the association remained unchanged. Individuals with genotype TA (rs4291) had lower odds of achieving blood pressure control in comparison with genotype AA (rs4291). However, the genotype GG (rs2239050) initially demonstrated an independent and significant association with controlled hypertension in response to amlodipine treatment in the crude logistic regression model analysis. After adjusting for confounding factors (drug regime), the effect was lost. After Bonferroni correction, TA (rs4291) remained significant with a *p*-value = 0.014 (Table 5).

## 4. Discussion

There is a paucity of existing literature on the ethnically and genetically diverse population of South Africa in relation to pharmacogenomics-based anti-hypertensive therapy. In the current study, we describe SNPs in amlodipine-associated genes and assess their correlation with blood pressure control among South African adults with hypertension.

The current study examined thirteen SNPs associated with amlodipine in 304 hypertensive individuals belonging to the indigenous Nguni tribe (Swati, Xhosa and Zulu) of South Africa. Out of thirteen, only five SNPs (rs1042713, rs10494366, rs2239050, rs2246709 and rs4291) were detected among the Xhosa population. This may be due to the small number of participants that were included in the study. This is not surprising as studies with a larger sample size demonstrate high statistical power and can detect smaller associations. Furthermore, the variant alleles of these SNPs were not detected among the Zulu and Swati tribal groups. Of note, the Swati tribal group is a minority in South Africa. However, failure to detect the selected SNPs among the Zulus is surprising, as the ratio of Xhosas to Zulus in the country is 1:1. The differences observed in the SNPs detected highlights the genetic differences that exist between the two indigenous tribes that might have been brought by demographic and evolutionary events, including admixture. 

The minor allele frequencies displayed by the Xhosa population were different from those observed in other African and global populations, including the Luhya (Kenya), the Yoruba (Nigeria and Benin), African American (United States of America), British (Great Britain), South Asian and Mexican (California, USA) people [28]. African populations, particularly those located in Southern Africa, are underrepresented in genomic studies [29]. The data presented in this study will help bridge the knowledge gap that exists and possibly contribute towards building an African-specific genomic database that could be utilized in personalised medicine. Furthermore, our study highlighted the diversity that exists among indigenous black South Africans. These differences could be used in predicting patients into responders and non-responders to amlodipine. To the best of our knowledge, this is the first study to detect and report all five SNPs in one of the indigenous ethnic groups of South Africa. Given the wide variations in different tribal groups in South Africa, more studies are, therefore, recommended to further expand the frontiers of pharmacogenomics in the country. Furthermore, future studies should put emphasis on the Zulus and the Swatis, as they form part of the dominant ethnic groups in the country. The panel of SNPs for future studies should put emphasis on the variants highlighted in this study and include other relevant SNPs, including those found in genes that encode drug transporter proteins, such as rs1045642 (ATP-binding cassette subfamily B member 1) and G2677T/C3435T (Multidrug-Resistance Protein 1) as they were previously associated with variable blood pressure response to amlodipine [27].

We found no association between blood pressure response to amlodipine and the genotypes or the alleles of rs1042713 (ADRB2) and rs2246706 (CYP3A5). In contrast, the G allele of rs1042713 (ADRB2) was significantly higher among Northern Han Chines individuals with essential hypertension [30]. It was further demonstrated that hypertensive patients with the AA genotype of rs2246706 (CYP3A5) may have a decreased likelihood of reaching a target mean arterial pressure, in comparison to carriers of the AG and GG genotypes [17]. The CYP3A5 gene plays an important role in amlodipine metabolism, whilst the ADRB2 gene encodes for a primary adrenergic receptor that causes vasodilation in humans [31,32]. Both SNPs showed no association with blood pressure response to amlodipine in the current study cohort and previous findings suggest that both polymorphisms may be of relevance in amlodipine pharmacogenomics [31,32,33]. Thus, our study does not allow any definitive conclusion and the clinical use of both CYP3A4 and ADRB2 SNPs for personalized amlodipine treatment regimens should be further explored in a larger hypertensive cohort of South African origin.

We found no association between genotypes and the alleles of NOS1AP rs10494366 and CACNA1C rs2239050 with blood pressure response to amlodipine. These findings are surprisingly different from previous reports. For instance, a study conducted among Caucasian patients showed that the GG genotype of rs2239050 was independently associated with an improved amlodipine treatment outcome rate of 52% [17]. Similar effects were observed among Chinese individuals [34]. Further, the genotype GG was associated with improved blood pressure response to amlodipine among Caucasian patients who were also carriers of the GG genotype of CACNA1C rs2238032, suggesting a possible SNP–SNP interaction [11]. The disparities observed in the present study and previous studies may be due to ethnic differences, different sample sizes, as well as the interference of other anti-hypertensive drugs that might have been prescribed to the patients. On the other hand, there is no record of the direct association of NOS1AP rs10494366 with blood pressure response to amlodipine. However, previous association studies suggest that this SNP may be associated with a higher risk of all causes of mortality among Caucasian participants on amlodipine therapy. The clinical relevance of CACNA1C and NOS1AP SNPs with regards to amlodipine therapy among people of African origin remains unknown. Thus, more studies need to be conducted to establish the relationship between these SNPs and blood pressure response to amlodipine among Africans.

Xhosa carriers of the TA genotype of ACE rs4291 were less likely to exhibit controlled blood pressure in response to amlodipine therapy. There is no record of the direct association of this polymorphism with blood pressure response to Amlodipine in the literature. However, the TA genotype was previously associated with decreased fasting plasma glucose levels among hypertensive patients undergoing amlodipine treatment [22]. This polymorphism occurs in a gene that is an important component in the renin-angiotensin-aldosterone system (RAAS), which acts as a key regulator of electrolyte balance [35]. As a result, the ACE gene is a good candidate for studying the pathophysiology of hypertension [9]. If the concept of precision medicine is to be realized, the functional effect of rs4291 needs to be further explored among a larger cohort that completely represents the South African Xhosa tribe. Further, future studies with a larger and more representative sample of the Zulu and Swati population should build on the current study to further elucidate the future role of rs4291 in pharmacogenomics-based amlodipine therapy. 

### Strength and Limitations of the Study

This is the first study to detect SNPs associated with amlodipine among indigenous South African ethnic groups. In addition, this study reports an association between the TA genotype of rs4291 and blood pressure control with amlodipine among the Xhosa tribe of a South African cohort. However, some limitations of the study cannot be ignored, including the small sample size, lack of information on the dosing of anti-hypertensive drugs and adherence to treatment and other lifestyle measures. These factors could have impacted on the extent of blood pressure control in the cohort. Further, the authors acknowledge the fewer samples of individuals on amlodipine monotherapy. This is largely due the standard of hypertension treatment in South Africa, where patients are initiated on thiazide diuretics upon diagnosis and other anti-hypertensive drugs are introduced as add-on drugs for patients who do not respond adequately to thiazide monotherapy [36]. Thus, we could not avoid interference from other anti-hypertensive drugs. As such, these findings are generalizable only to the population of individuals living within the two regions (Eastern Cape and Mpumalanga) and similar settings in the country. Additionally, rs10427139 and rs4291 were previously associated with variable responses to other anti-hypertensive drugs (atenolol and enalapril) [27]. Therefore, the inclusion of the two variants in our genotyping panel may have significantly biased our findings. Furthermore, there was no variation observed at the chosen loci among the Zulu and Swati ethnic groups. It should also be noted that this study purposively selected three populous tribal groups in the country; therefore, more studies are needed among other ethnic groups in South Africa.

## 5. Conclusions

This study reports the detection of five SNPs in amlodipine-associated genes (rs2239050, rs2246709, rs4291, rs1042713 and rs10494366) among the indigenous Xhosa-speaking ethnic group of South Africa. In addition, the TA genotype of rs4291 was associated with blood pressure response to amlodipine treatment among the Xhosa cohort. Findings of the study highlight the relevance of comprehensively characterizing highly diverse populations, particularly those of African origin, to facilitate pharmacogenomics-based anti-hypertensive treatment. Additionally, these findings might open doors for more pharmacogenomics studies, which could inform innovations to personalized anti-hypertensive treatment in the ethnically diverse population of South Africa. Furthermore, our findings laid a foundation for building an African-specific panel of SNPs that could be used to identify responders and non-responders to amlodipine as well as other anti-hypertensive drugs, thus, providing the right treatment to the right patient at the right time.

## Figures and Tables

**Table 1 genes-13-01394-t001:** Selected amlodipine variants used in the design of multiplex MassARRAY panels (*n* = 13).

SNP	GENE	Level of Evidence	Reference
rs1045642	*ABCB1*	3	www.pharmgkb.org (accessed on 28 February 2020).
rs10494366	*NOS1AP*	3	www.pharmgkb.org (accessed on 28 February 2020).
rs11122576	*AGT*	3	www.pharmgkb.org (accessed on 28 February 2020).
rs12143842	*AGT*	3	www.pharmgkb.org (accessed on 28 February 2020).
rs1799752	*ACE*	3	www.pharmgkb.org (accessed on 28 February 2020).
rs2246709	*CYP3A4*	3	www.pharmgkb.org (accessed on 28 February 2020).
rs2740574	*CYP3A4*	3	www.pharmgkb.org (accessed on 28 February 2020 ).
rs4291	*ACE*	3	www.pharmgkb.org (accessed on28 February 2020 ).
rs2032582	*ABCB1*	3	www.pharmgkb.org (accessed on 28 February 2020).
rs1042713	*ADBR2*	3	www.pharmgkb.org (accessed on 28 February 2020).
rs10494366	*NOS1AP*	3	www.pharmgkb.org (accessed on 28 February 2020).
rs2239050	*CACNA1C*	3	www.pharmgkb.org (accessed on 28 February 2020).
rs2238032	*CACNA1C*	3	www.pharmgkb.org (accessed on 28 February 2020).

**Table 2 genes-13-01394-t002:** General characteristics of study participants disaggregated by sex (*n* = 304).

Variables	All Participants (*n*; %)	Males (*n*; %)	Females (*n*; %)
All	304 (100%)	71 (23.3%)	233 (76.6)
Age (Years)			
18–25	1 (0.3)	-	1 (0.4)
26–35	9 (3,0)	5 (7,0)	4 (1.7)
36–45	23 (7.6)	3 (4.2)	20 (8.6)
46–55	65 (21.4)	16 (22.5)	49 (21.0)
56–65	97 (31.9)	24 (33.8)	73 (31.3)
≥66	109 (35.9)	23 (32.4)	86 (36.9)
Ethnicity			
Zulu	139 (45.7)	25 (35.1)	114 (48.9)
Swati	43 (14.1)	6 (8.5)	37 (15.9)
Xhosa	122 (40.1)	40 (56.3)	82 (35.2)
Smoking status			
Never Smoked	259 (85.2)	42 (59.2)	217 (93.1)
Ever Smoked	45 (14.8)	29 (40.2)	16 (6.9)
Salt intake			
Low-Moderate	237 (78.0)	52 (73.2)	185 (79.4)
Increased	67 (22.0)	19 (26.8)	48 (20.6)
Blood Pressure			
<140/90 mmHg	145 (47.7)	25 (35.2)	120 (51.5)
≥140/90 mmHg	159 (52.3)	46 (64.8)	113 (48.5)
Drug Regime			
Amlodipine Alone	13 (4.3)	4 (5.6)	9 (3.9)
Amlodipine + 1 Drug	113 (37.2)	25 (35.2)	88 (37.8)
Amlodipine + 2 Drugs	152 (50.0)	36 (50.7)	116 (49.8)
Amlodipine + 3 Drugs	26 (8.65)	6 (8.5)	20 (8.6)

Drugs used in combination with Amlodipine: Hydrochlorothiazide, Enalapril, Atenolol.

**Table 3 genes-13-01394-t003:** Distribution patterns of selected single nucleotide polymorphisms (SNPs).

SNP	Gene	Ethnic Groups			Gender		Age		
		Zulu (*n*; %)	Swati (*n*; %)	Xhosa (*n*; %)	Male (*n*; %)	Female (*n*; %)	<55 Years	55–65 Years	>65 Years
**All**		139 (45.7)	43 (14.1%)	122 (40.1%)	40 (32.8%)	82 (67.2%)	24 (19.7%)	55 (45.1%)	43 (35.2%)
**rs1042713**	*ADBR2*								
Yes		-	-	97 (79.5)	34 (85.0)	63 (76.8)	19 (79.2)	43 (78.2)	35 (81.4)
No		139 (100)	43 (100)	25 (20.5)	6 (15.0)	19 (23.2)	5(20.8)	12 (21.8)	8 (18.6)
**rs10494366**	*NOS1AP*								
Yes		-	-	119 (97.5)	39 (97.5)	80 (97.6)	23 (95.8)	54 (98.2)	42 (97.7)
No		139 (100)	43 (100)	3 (2.5)	1 (2.5)	2 (2.4)	1 (4.2)	1 (1.2)	1 (2.3)
**rs2239050**	*CACNA1C*								
Yes		-	-	122 (100)	40 (100)	82 (100)	24 (100)	55 (100)	43 (100)
No		139 (100)	43 (100)	-	-	-	-	-	-
**rs2246709**	*CYP3A4*								
Yes		-	-	122 (100)	40 (100)	82 (100)	24 (100)	55 (100)	43 (100)
No		139 (100)	43 (100)	-	-	-	-	-	-
**rs4291**	*ACE*								
Yes		-	-	122 (100)	40 (100)	82 (100)	24 (100)	55 (100)	43 (100)
No		139 (100)	43 (100)				-	-	-

**Table 4 genes-13-01394-t004:** Minor allele frequency distribution across different population groups.

SNP	Nucleotide Substitution	Feature	Minor Allele Frequency (%)						
			Xhosa	Yoruba	Luhya	African American	Mexican	British	South Asian
**rs1042713**	G > A	Missense	45.9	88.0	78.8	87.7	85.9	60.4	80.7
**rs10494366**	G > T	Intron	35.3	88.0	86.4	77.9	57.8	50.0	60.3
**rs2239050**	C > G	Intron	28.7	87.5	85.9	83.6	72.7	53.8	74.5
**rs2246709**	A > G	Intron	51.6	13.4	12.6	14.8	14.9	20.3	14.0
**rs4291**	T > A	Regulatory	34.4	22.7	23.2	39.3	41.4	44.0	38.4

**Table 5 genes-13-01394-t005:** Adjusted and unadjusted logistic regression models showing genotypes and alleles associated with blood pressure control.

SNP	Controlled Hypertension (*n*; %)	Uncontrolled Hypertension (*n*; %)	Unadjusted Odds Ratios (95% CI)	*p*-Value	Adjusted Odds Ratios (95% CI)	*p*-Value	Bonferroni Adjusted *p*-Value
All	20 (16.4%)	102 (83.6%)					
**rs1042713**							
Genotypes							
GG	4 (15.4)	22 (84.6)	1		1		
GA	10 (15.2)	56 (84.8)	1.29 (0.32–5.21)	0.718	0.68 (0.19–2.43)	0.559	
AA	6 (20.0)	24 (80.0)	1.05 (0.15–7.26	0.809	0.76 (0.13–4.38)	0.768	
Alleles							
G	18 (15.3)	100 (84.7)	1		1		
A	22 (17.5)	104 (82.5)	1.04 (0.47–2.26)	0.928	1.12 (0.47–2.63)	0.787	
**rs10494366**							
Genotypes							
TT	13 (18.1)	59 (81.9)	1		1		
GT	3 (13.0)	20 (87.0)	0.77 (0.22–2.62)	0.958	0.59 (0.12–2.91)	0.520	
GG	4 (14.8)	23 (85.2)	0.74 (0.190–2.90)	0.669	0.56 (0.13–2.29)	0.426	
Alleles							
T	29 (17.4)	138 (82.6)	1		1		
G	11 (14.3)	66 (85.7)	0.53 (0.25–1.16)	0.113	0.58 (0.25–1.34)	0.208	
**rs2239050**							
Genotypes							
CC	11 (19.0)	47 (81.0)	1		1		
CG	5 (8.8)	52 (91.2)	0.41 (0.132–1.54)	0.122	0.36 (0.10–1.26)	0.111	
GG	4 (57.1)	3 (42.9)	**5.69 (1.11–29.21)**	**0.003**	2.49 (0.51–12.13)	0.257	0.128
Alleles							
C	27 (15.6)	146 (84.4)	1		1		
G	13 (18.3)	58 (81.7)	1.07 (0.51–2.26)	0.841	0.76 (0.29–1.99)	0.583	
**rs2246709**							
Genotypes							
GG	3 (27.3)	8 (72.7)	1		1		
AG	8 (16.7)	40 (83.3)	0.37 (0.07–1.86)	0.183	0.44 (0.05–3.28)	0.423	
AA	9 (14.3)	54 (85.7)	0.24 (0.04–1.23)	0.730	0.32 (0.04–2.23)	0.267	
Alleles							
G	14 (20.0)	56 (80.0)	1		1		
A	26 (14.9)	148 (85.1)	0.73 (0.37–1.44)	0.359	0.72 (0.32–1.59)	0.420	
**rs4291**							
Genotypes							
AA	11 (21.6)	40 (78.4)	1		1		
**TA**	4 (6.8)	55 (93.2)	**0.26 (0.07–0.89)**	**0.003**	**0.23 (0.06–0.85)**	**0.027**	**0.013**
TT	5 (41.7)	7 (58.3)	2.59 (0.68–9.79)	0.158	3.65 (0.86–15.48)	0.078	
Alleles							
A	26 (16.1)	135 (83.9)	1		1		
T	14 (16.9)	69 (83.1)	1.20 (0.61–2.39)	0.596	1.27 (0.54–3.01)	0.571	

CI: confidence interval.

## Data Availability

The data presented in this study is available from the corresponding author upon reasonable response.

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
