# Peer review of "Single Nucleotide Polymorphisms in Amlodipine-Associated Genes and Their Correlation with Blood Pressure Control among South African Adults with Hypertension"

_genes, 2022, doi:10.3390/genes13081394_

Round 1

Reviewer 1 Report

The authors studied Amlodipine associated genes in South African adults with Hypertension and found important relationships. I have some concern as follows:

Results

Table 2. The zero before single numbers make confusion. It is better to write without zero like 1 instead of 01. Moreover this type of correction is necessary for all tables.

Table 2. It is better not to put the parentheses together with the number, but to leave a space after the number. It is also required for all tables.

Discussion

“Although, many advances have been made in the field of hypertension therapeutics,  inter-individual variability in response to the various classes of anti-hypertensive drugs have been reported (13).” This is given in the Introduction part and  this kind of repeat is unnecessary for Discussion part. One of them must be deleted.

“single nucleotide polymorphisms (SNPs)” this is the third time. It should be only in the first part of the Introduction. You have to write only SPNs fort he second and third time.

Author Response

Dear Reviewer,

Thank you for taking the time to review our manuscript on “Single Nucleotide Polymorphisms in Amlodipine associated genes and their correlation with blood pressure control among South African adults with Hypertension”. The suggestions that you have offered  have been immensely helpful, and we grateful for  your insightful comments.

All the suggested changes have been made and are highlighted throughout the manuscript. All three authors have read and approved the revised manuscript.

Thank you.

Dr Charity Masilela (for authors)

Reviewer 2 Report

Masilela and colleagues checked 12 amlodipine single nucleotide polymorphisms among 304 hypertensive patients belonging to the indigenous South African Swati, Xhosa and Zulu populations. Out of these patients, only n=13 (4.3%) were on amlodipine monotherapy, while 37% were on two, 50% on three and 9% on four antihypertensive drugs. 52% of participants had insufficiently controlled hypertension under their current therapy.

Out of the 12 tested SNPs, only 5 were detected, and only among the Xhosa participants. Surprisingly, no single SNP was detected among the Swati and Zulu tribal groups. Out of the 5 SNPs detected among the Xhosa participants, only one, i.e. the TA genotype of the rs4291, was significantly associated with uncontrolled hypertension.

The study theme is scientifically interesting. Its potential clinical impact might reside in a better care of the Xhosa population, while its extrapolation to a broader context might be problematic and very limited. Nevertheless, a better understanding of the pharmacogenetic sensitivity to antihypertensive drugs is potentially useful worldwide.

Globally, the study design is in principle appropriate. However, given that only approximately 4% of the participants were on amlodipine monotherapy (page 6 and Table 1, see specific comment below) I am afraid that the studied population might not be appropriate to answer the study question of better understanding mechanisms implicated in blood pressure response to amlodipine. Do Authors have proofs that there is no role for the investigated genes in the absorption, distribution, metabolism, elimination and concentration-effect relationship of the other allowed anti-hypertensive drugs (like enalapril, atenolol, hydrochlorothiazide ...)?

The analysis appears to have been carefully performed, and it is well reported.

The Introduction is too long and should be shortened by at least 30-40% of its current length. The Methods and Results are well reported.

The Discussion, however, needs to be completely rewritten. Actually, it is currently too long (and needs to be shortened by at least 50% of its current length), not well-structured and, at the end, even confusing. For example, Authors seem to put their own results into question (page 10, lines 67-71). Furthermore, Authors are victims of the (sadly quite common) temptation of proving to the Editor, Reviewers and Readers that they had read the literature. I congratulate with them for their knowledge of the literature, but this is actually not the point. A Discussion of an original article is not a narrative review. It is implicit to any research study (and ethical submission) that Authors know the relevant literature. In an original study (as opposed to a narrative review), the aim of a Discussion is to discuss and interpret the results of its own study, and to discuss their impact on clinical care and future research. Unfortunately, Authors cite a lot of other works (even addressing endometrial cancer and metformin! Page 10, lines 100-101 and 105), but forget to interpret several of their own results. Specifically, Authors must address and discuss (at least) following points:

1)    Why only 5 out of 12 tested SNPs were detected?

2)    Why only in the Xhosa population?

3)    Why the other studied populations did not show any of the 12 screened SNPs?

4)    Should other SNPs be included in screening studies? Which ones? Why?

5)    What about Swati and Zulu? What could be done to enravel their specificities and to improve their therapy? Proposals?

6)    Why only 1 out of 5 SNP was associated with “amlodipine sensitivity” in the Xhosa population?

7)    What the impact of the co-medications? Could the sensitivity to the other antihypertensive drugs have significantly biased the results? What about the SNPs among the 4% being on amlodipine monotherapy as compared to the other participants? Is something known on the impact of the 5 detected SNPs on the pharmacokinetics and pharmacodynamics of enalapril, atenolol and hydrochlorothiazide?

Furthermore, if Authors really need to cite other works and they strongly feel that this adds to the understanding of their own results, they must discuss and explain (at least temptatively) the differences in the results obtained by others as compared to their owns, and also explain why their study is better/more reliable/more representative as compared to the previous ones.

Following specific comments must be addressed.

Page 1, Abstract: substitute “MAF” with “minor allele frequency” throughout the abstract.

Page 2: “a prevalence of up to” -> “a prevalence of arterial hypertension of up to”

Page 2: “Despite the high prevalence, the rate of uncontrolled hypertension is high among South African adults [6].” -> “Arterial hypertension is highly prevalent among South African adults, and unfortunately the rate of uncontrolled hypertension is high as well [6]”.

Page 2: “during drug selection and dosing in to ensure optimal” -> “during both drug selection and dosing to ensure optimal”

Page 3: “Although, these findings have” -> “Although these findings have”

Page 3 (“More studies based on ethnically diverse populations are needed in order to improve and guide treatment strategies for African specific populations.”): I understand what you would like to say, but what you are actually writing is incorrect, and there is currently an inconsistency: do you focus on “ethnical diversity” or on “specific populations”? Either you want to be comprehensive (and cover all the diversity) or you wish to look at a specific population.

Proposal: “In order to improve the treatment of arterial hypertension in South-Africa, studies on its ethnically diverse sub-populations are needed.”

Page 4 (“active if participants engaged in exercise leading to an increase in heart and respiratory rate, such as gardening…”): with which frequency? Was this defined? E.g. 1x/day, 3x/week, 1x/week, 1x/month, …?

Throughout the manuscript (for example page 6: “A total of 304 patients participated in the study, of which 23.36% (n=71) were males and 76.64% (n=233) were females. The majority of the study participants were older than 45 years (n=271), 85.19% (n=259) had

never smoked, 77.96% (n=237) used low-moderate salt intake and 53.31% (n=159) had blood pressure ≥140/90 mmHg. Overall, 4.28% (n=13) of the participants were on amlodipine monotherapy (Table 1)”): use only max. 1 decimal digit for %. Ideally, if they are ≥10% no decimal digit at all, and if they are <10% one decimal digit. For example here: “23.36%” -> “23%”, respectively “4.28%” -> “4.3%”.

Table 2: same comment regarding significant digits for % values.

Page 7: same comment.

Table 3: same comment.

Table 4: same comment.

Table 5: same comment.

Please also see general comment above: Page 6 (“Overall, 4.28% (n=13) of the participants were on amlodipine monotherapy (Table 1)”): i.e. less than 1 out of 20. Is this population appropriate to answer your study question? Is there no role for the investigated genes in the absorption, distribution, metabolism, elimination and concentration-effect relationship of the other allowed anti-hypertensive drugs (like enalapril, atenolol, hydrochlorothiazide ...)? Is this design appropriate?

Table 2: in fact, approximately half was on Amlodipine + 2 drugs, almost 10% on even 3 drugs, and >95% on any combination therapy… (this just adds to my previous comment)

Table 3:

-       for what does “dbSNP” (i.e. db) stand for? …??... single nucleotide polymorphisms? If you really need to use abbreviations (by the way, why?), then define them.

-       Eliminate the “No raws”, and just provide the data for the “yes” rows (the “no” rows must be, by definition, “No raw” = (100% - “Yes raw”)).

Table 5: same comment:

Page 8, line 2: “MAF” -> “minor allele frequency”

Page 8, line 9: “34.4%) showed a higher MAF; however,” ->34.4%) showed a higher minor allele frequency; however,

Table 5:

-       Same comment as for Table 3: for what does “dbSNP” (i.e. db) stand for? …??... single nucleotide polymorphisms? If you really need to use abbreviations (by the way, why?), then define them.

-       “HPT” -> “HTN” (if you really feel that there is any need, or any advantage, of writing HTN instead of hypertension: there is sufficient place here!)

Page 9, bottom: “Genes 2021”: we are now in June 2022. The Editorial office should correct this mistake.

Table 5, bottom legend:

1) I do not find these symbols (* and **) in the table, sorry!

2) what did you compare? controlled vs uncontrolled (same row) or different genes (different rows) or different populations?

Page 9, line 31: “have been reported” -> “has been reported”

Page 9, line 34: “and assesses” -> “and assess” (the subject is “we”, line 33)

Page 9, line 39: “population. However, the variant alleles of” -> “population. The variant alleles of”

Page 9, line 49: “stratifying” -> “predicting” (this stresses and makes clear the interest of your study: to be able to predict their expected response to amlodipine before starting the treatment, while it is already now possible to stratify the patients, based on their clinical response over time...)

Page 9, lines 50-51: “patient to responders and non-responder to anti-hypertensive drugs including amlodipine.” -> “patients into responders and non-responder to amlodipine.”

Page 10, lines 55-71:

1)    Reduce this paragraph by 40-50% of its current length.

2)    Start with: “Surprisingly, some of our results were different from previous literature. First, …” and so on.

Page 10, lines 72-95: please see general comment above.

1)    Reduce by at least 50% of the current length.

2)    Discuss and interpret YOUR results. This is an original study, not a narrative review of previous literature. In particular, you must address (either explain or provide one or more hypothesis of explanation):

a)    Why only 5 out of 12 tested SNPs were detected?

b)    Why only in the Xhosa population?

c)    Why the other studied populations did not show any of the 12 screened SNPs?

d)    Should other SNPs be included in screening studies? Which ones? Why?

e)    What about Swati and Zulu? What could be done to enravel their specificities and to improve their therapy? Proposals?

f)     Why only 1 out of 5 SNP was associated with “amlodipine sensitivity” in the Xhosa population?

Page 10, line 66: “ADRB2 gene encode for” -> “ADRB2 gene encodes for”

Page 10, lines 67-71: Please also see general comment above. Sorry, I am getting confused here. Do you believe in your own results or not? If you contradict your own results, you deliver the message that you do not trust your own study, discrediting it completely. How can the reviewers, the editor and the readership believe in a study in which the Authors themselves do not believe?

Page 10, lines 81-84: and different ethnicities! Is there a polygenic pattern? Probably yes, because the general adult population mainly suffers from idiopathic arterial hypertension rather than from monogenic forms of hypertension.

Page 10, lines 87-89: did you perform ECG in your patients? Did you measure QTc? If yes, please provide this data. If not, delete this sentence.

Page 10, lines 100-101 (“Also, rs4291 was associated with increased plasma ACE activity in endometrial cancer (21)”): irrelevant to this study. Please delete.

Page 10, line 103: “electrolyte imbalance (36).” -> “electrolyte balance (36).”

Page 10, line 105: “complicated hypertension and pharmacodynamics of metformin (9).” -> “complicated hypertension (9).”

Page 11, line 111: “with amlodipine among a South African cohort.” -> “with amlodipine among the Xhosa tribe in South Africa.”

Page 11, lines 112-114: and especially the huge bias of co-medication with further antihypertensive drugs!

Page 11, lines 116-119: This is surprising (and different from international guidelines). The priority order should rather be 1) ACEi, 2) CCB, 3) bB, 4) diuretics. The fact that you studied a population treated against international recommendations might quite significantly bias the results. Importantly, they cannot be extrapolated to any other country and their impact remains only regional/national. Obviously, the interest of your study still exists in showing ethnic diversity in genes potentially impacting drug pharmacodynamics.

Author Response

(The authors gave the same response as above.)

Reviewer 3 Report

Overall a nice paper. Can you summarise the important learning points of this study and how it can help hypertensive patients on Amlodipine? Otherwise the scientific rigor of this study is robust

Author Response

(The authors gave the same response as above.)

Round 2

Reviewer 2 Report

The revised version has clearly improved. However, there are still several points, which need to be addressed.

I have two main concerns and several minor requests.

A) the bias represented by the confounding of co-medications limits the interpretation of the whole study and is the key limitation of this research. Therefore, it is still not sufficiently stressed and its impact is still not addressed (see for examples lines 136-146). I already tried to explain in my previous report that this key limitation might be mitigated by:

1) referring to any knowledge on the role of the investigated SNP's on the PK and PD of other anti-hypertensive drugs,

2) sensitivity analysis comparing all group / only amlodipine monotherapy / only combined anti-hypertensive regimen.

Authors should spend some time to work on both these points. Please note that this represents, in my eyes (i.e. as per opinion of this reviewer), condition for acceptance.

B) I already raised several questions, relating both to your methodological choices and to the interpretation of your results, which need to be addressed in the Discussion, but which are mainly still unaddressed (or insufficiently addressed). Authors need to develop some in-depth reflection on these point and deliver satisfactorily explanations.

1)    Why only 5 out of 12 tested SNPs were detected?

2)    Why only in the Xhosa population?

3)    Why the other studied populations did not show any of the 12 screened SNPs?

4)    Should other SNPs be included in screening studies? Which ones? Why?

5)    What about Swati and Zulu? What could be done to enravel their specificities and to improve their therapy? Proposals?

6)    Why only 1 out of 5 SNP was associated with “amlodipine sensitivity” in the Xhosa population?

7)    What the impact of the co-medications? Could the sensitivity to the other antihypertensive drugs have significantly biased the results? What about the SNPs among the 4% being on amlodipine monotherapy as compared to the other participants? Is something known on the impact of the 5 detected SNPs on the pharmacokinetics and pharmacodynamics of enalapril, atenolol and hydrochlorothiazide? (Point 7 is partly the repetition of main concern A above.)

8)    I also already stressed that the fact that you studied a population treated against international recommendations might quite significantly bias the results (the internationally recommended hierarchical order for idiopathic arterial hypertension is not to start with hydrochlorthiazide as the first anti-hypertensive drug, but rather: 1) ACEi, 2) CCB, 3) bB, 4) diuretics, of which hydrochlorothiazide is first choice. See current lines 137-139, respectively previous comment related to Page 11, lines 116-119 of the original submission). Importantly, this implies that your results cannot be extrapolated to any other country and their impact remains only regional/national. Obviously, the interest of your study still exists in showing ethnic diversity in genes potentially impacting drug pharmacodynamics. This limitation is still not listed in your revised discussion.

The submission can be considered only after satisfactorily working on these 2 issues.

Furthermore, following minor requests must also be integrated.

Page 1 of the pdf, Abstract:

-       “ to identify the significant associations” -> “ to identify significant associations”

-       “Variants, rs1042713” -> “Variants,rs1042713”

Page 2 of the pdf, Introduction

-       “Health Survey a prevalence of arterial” -> “Health Survey detected a prevalence of arterial”

-       adults [2]. Whereas” -> “adults [2], whereas”

-       “Additionally, studies conducted in KwaZulu Natal, Eastern Cape, Gauteng, and the Limpopo province of South Africa have reported a range of prevalence between 12.4% and 52.0% [1,3–5]”: delete

-       “[6]. A recent study reported an overall prevalence of uncontrolled hypertension of 56.83% among rural dwellers of the Mpumalanga province [7]. Whilst a 75.5% and 51.0% prevalence were reported in rural Eastern Cape and KwaZulu Natal, respectively [6,8]. As hypertension” -> “[6], with reported prevalenss between 50 and 76% [6-8]. As hypertension”

-       “The CACNA1C gene encodes for an alpha-1 subunit of a voltage-dependent calcium channel that mediates the influx of calcium ions into the cell upon membrane depolarization [16].” : delete

-       “Both ACE and AGT are central components of the renin–angiotensin system (RAS) that controls blood pressure by regulating the volume of fluids in the body [19] .”: delete

-       “Variant rs4291 of the ACE gene was strongly associated with the incidence of high blood pressure [9], however; its direct association with blood pressure response to amlodipine remains elusive. Similarly, African American carriers of the minor allele of rs11122576 of AGT gene who were undergoing amlodipine therapy showed a decreased risk of coronary heart diseases, however, no clear association with blood pressure in response to amlodipine has been established [20].”: delete

Page 3 of the pdf, Introduction

-       “Additionally, Beta-adrenergic receptor (ADRB2) and nitric oxide synthase-1-adaptor protein (NOS1AP) are part of the sympathetic and para-sympathetic nervous systems and are known to be involved in the pathophysiology of hypertension [21,22]. The minor allele G of rs10494366 of the NOS1AP gene was associated with an increase in cardiovascular mortality among Caucasian users of amlodipine [22]. Furthermore, patients with the AA genotype of rs1042713 (ADRB2) demonstrate poor efficacy of cardiovascular drugs including ACE-inhibitors [23]. Nevertheless, no direct association with blood pressure response to amlodipine has been established for both SNPs. More studies need to be conducted in order toto establish a clear association between these SNPs and blood pressure response to amlodipine.”: delete

-       “drugs in order toto improve drug efficacy among patients (24).” -> “drugs (24).”

Page 4 of the pdf, Methods

-       “meters squared” -> “squared meters”

-       “62 °C overnight Thereafter, DNA” -> “62 °C overnight. Thereafter, DNA”

Page 6 of the pdf, Results

-       23.36% -> 23.4% (not 23.3%)

-       77.96% -> 78.0% (not 77.9%)

-       4.28% -> 4.3%

Page 7 of the pdf, Results

-       79.51% -> 79.5%

-       76.82% -> 76.8%

-       78.18% -> 78.2%

-       97.56% -> 97.6%

-       98.18% -> 98.2%

Page 8, Lines 34-36: “In the current study, we describe we describe single nucleotide polymorphisms (SNPs) in amlodipine associated genes and assesses their correlation with blood pressure control among South African adults with hypertension”: delete

Page 11, lines 71-73: “In contrast, the G allele of rs1042713 (ADRB2) was significantly higher among Northern Han Chines individuals with essential hypertension (30).”: delete

Page 11, line 80: “Although both SNPs” -> “Both SNPs”

Page 11, lines 81-82: “cohort, previous findings suggest that both polymorphisms may be of relevance in amlodipine pharmacogenomics (31, 33)” -> “cohort, as opposed to previous findings (31, 33).”

Page 11, line 83: “Thus, the clinical use of” -> “Thus, our study does not allow any definitive conclusion and the clinical use of”

Page 11, line 98: “reference studies” -> “previous studies”

Page 11, line 100 – Page 12, line 104 (“On the other hand, there is no record of the direct association of NOS1AP rs10494366 with blood pressure response to amlodipine. However, previous association studies suggest that this SNP may be associated with a higher risk of all causes of mortality among Caucasian participants on amlodipine therapy.”): delete

Page 12, line 119: “component the renin” -> “component of the renin”

Page 12, lines 126-127: “the future role” -> “the role”

Page 12, lines 136-137: “Also, the authors acknowledge the fewer 136 samples of individuals on amlodipine monotherapy. This is largerly due…” -> “Also, the authors acknowledge that only  4% of participants were on amlodipine monotherapy. This represents a major limitation. In fact, the sensitivity to the other antihypertensive drugs may have significantly biased the results. This was largely due…

Lines 136-144: see general comment above. To try to mitigate the huge bias, Authors must: 1) summarize any evidence of the role of the investigate SNPs in the PK and PD of the other anti-hypertensives used by their study cohort, 2) perform sensitivity analyses.

Authors should note that the 2 major concerns listed above, as well as the specific requests regarding deleting some parts of the manuscript (in order to improve its readability and appeal to the readership and to increase its potential to be cited in future works) are mandatory requests.

Author Response

Dear reviewer 2,

Thank you for taking the time to review our manuscript on “Single Nucleotide Polymorphisms in Amlodipine associated genes and their correlation with blood pressure control among South African adults with Hypertension”.  We are grateful for the comments and insight that you have  provided.  Attached is our point by point response to all the comments.

Thank you.

Dr Charity Masilela (for authors)
